# Exploring the Bio-Functional Effect of Single Nucleotide Polymorphisms in the Promoter Region of the TNFSF4, CD28, and PDCD1 Genes

**DOI:** 10.3390/jcm12062157

**Published:** 2023-03-10

**Authors:** Ding-Ping Chen, Ying-Hao Wen, Wei-Ting Wang, Wei-Tzu Lin

**Affiliations:** 1Department of Laboratory Medicine, Linkou Chang Gung Memorial Hospital at Linkou, Taoyuan City 333, Taiwan; 2Department of Medical Biotechnology and Laboratory Science, College of Medicine, Chang Gung University at Linkou, Taoyuan City 333, Taiwan; 3Graduate Institute of Clinical Medical Sciences, College of Medicine, Chang Gung University at Linkou, Taoyuan City 333, Taiwan; 4School of Medicine, National Tsing Hua University, Hsinchu 30013, Taiwan; 5School of Medicine, Chang Gung University at Linkou, Taoyuan City 333, Taiwan

**Keywords:** bio-function, single nucleotide polymorphism (SNP), promoter, TNFSF4, PDCD1, CD28, transcriptional activity

## Abstract

In a prior study, we discovered that hematopoietic stem cell transplantation (HSCT) and/or autoimmune diseases, such as systemic lupus erythematosus, were associated with the rs1234314 C/G and rs45454293 C/T polymorphisms of *TNFSF4*, the rs5839828 C > del and rs36084323 C > T polymorphisms of *PDCD1*, and the rs28541784C/T, rs200353921A/T, rs3181096C/T, and rs3181098 G/A polymorphisms of *CD28*. However, the association does not imply causation. These single nucleotide polymorphisms (SNPs) are all located in the promoter region of these genes, so we used the dual-luminescence reporter assay to explore the effect of single nucleotide polymorphisms (SNPs) on transcriptional activity. For each promoter–reporter with a single SNP mutation, more than 10 independent experiments were carried out, and the difference in transcription activity was compared using one-way ANOVA and Tukey’s honestly significant difference test. The results showed that the G-allele of rs1234314 had 0.32 ± 0.09 times the average amount of relative light units (RLU) compared to the C-allele (*p* = 0.003), the T-allele of rs45454293 had 4.63 ± 0.92 times the average amount of RLU compared to the C-allele (*p* < 0.001), the del-allele of rs5839828 had 1.37 ± 0.24 times the average amount of RLU compared to the G-allele (*p* < 0.001), and the T-allele of rs36084323 had 0.68 ± 0.07 times the average amount of RLU compared to the C-allele (*p* < 0.001). The CD28 SNPs studied here did not affect transcriptional activity. In conclusion, the findings of this study could only confirm that the SNP had a bio-functional effect on gene expression levels. According to the findings, several SNPs in the same gene have bio-functions that affect transcriptional activity. However, some increase transcriptional activity while others decrease it. Consequently, we inferred that the final protein level should be the integration result of the co-regulation of all the SNPs with the effect on transcriptional activity.

## 1. Introduction

Even though hematopoietic stem cell transplantation (HSCT) has become the gold standard therapy for hematological diseases and cancer, post-HSCT complications limit its therapeutic potential. Although standardization of human leukocyte antigen (HLA) matching between donor and patient before transplantation has greatly improved HSCT outcomes, the mortality rate caused by many complications after transplantation, such as graft-versus-host disease (GVHD), infection, and disease relapse, remains significantly higher than that of the general population [1,2]. Accumulating evidence supports that *non-HLA* gene variations are an important cause of post-HSCT outcomes [3,4,5]. The immune system’s role is largely responsible for the immune system’s reaction to alloantigens and infection complications after HSCT. Co-stimulatory molecules are one of the most important components in immune regulation, and they are likely to be affected by gene polymorphisms [5,6]. It is known that T-cell activation requires two necessary signals. The first signal is produced when the T-cell receptor interacts with the antigen peptide presented by the major histocompatibility complex molecule on the antigen-presenting cells (APCs), and the second is a positive regulation signal of T-cell activation produced by the interaction between the receptor expressed on the surface of T cells and its ligand expressed on the APCs [7,8,9,10]. The second signal will reduce T-cell proliferation and cytokine production, promote T-cell dysfunction or apoptosis, and activate regulatory T cells [11].

Imbalance of the co-stimulatory system is one of the immune escape mechanisms in blood cancers. Many studies [12,13,14,15] have shown a link between HSCT and co-stimulatory molecules, such as tumor necrosis factor superfamily 4 (*TNFSF4*), programmed cell death protein 1 (*PDCD1*; PD-1; CD279), and *CD28*. These genes were also associated with the development of autoimmune diseases and cancers [16]. According to some studies, CD28 and PDCD1 play an important role in the immune system and transplantation [8,17,18,19]. Furthermore, genetic variations, such as SNPs in *HLA* and *non-HLA* genes, were linked to HSCT success or failure in various ethnic groups [20].

Systemic lupus erythematosus (SLE) is an autoimmune disease (AD) characterized by the production of autoantibodies. Its pathogenesis is still unclear. Several studies have shown that the overactivation of autoreactive T cells is the primary cause of Alzheimer’s disease [21,22]. Although the current study found that the *HLA* gene usually had the strongest correlation with AD [23], other genes located out of the *HLA* region may also be risk factors for AD. SNPs in genes encoding T-cell and B-cell function-related proteins were identified as SLE susceptibility loci. Among them, the *CD28* gene, which continuously expresses on T cells and provides the second stimulation signal to promote T-cell activation and make it aggressive after binding with CD80/CD86 on APCs, is the most important [24,25]. *PDCD1* is a type of immune checkpoint. When PD-1 protein binds to its ligand on APCs, it activates an immunoreceptor tyrosine-based inhibitory motif on the cytoplasmic tail of PD-1, thus inhibiting T-cell activation [26]. In addition, *TNFSF4* (OX40L) interacts with its receptor (OX40) and can also provide signals to promote T-cell activation, with previous research showing that OX40L is capable of stimulating T-cell response as well as promoting the pathogenesis of SLE [27].

We previously discussed the association between SNPs of the *CD28*, *TNFSF4*, and *PDCD1* genes and the outcomes of post-cord blood transplantation (CBT) and HSCT and the development of SLE [28,29,30], finding that many SNPs located in the promoter region had statistically significant differences. However, the association does not imply causation. As a result, fluorescence analysis for SNPs in the promoter regions of the *TNFSF4*, *PDCD1*, and *CD28* genes was performed to investigate the impact of these SNP changes on transcription activity.

## 2. Materials and Methods

### 2.1. SNP Selection

In a previous study, the promoter regions of *CD28*, *PDCD1*, and *TNFSF4* were amplified to investigate the relationship between SNPs in these regions and HSCT, CBT, and SLE. Of interest were the *TNFSF4* polymorphisms rs1234314 C/G and rs45454293 C/T, *PDCD1* polymorphisms rs5839828 C/del and rs36084323 C/T, and *CD28* polymorphisms rs28541784 C/T, rs200353921 A/T, rs3181096 C/T, and rs3181098 G/A. Thus, these promoter SNPs were selected to analyze the effect on transcription activity. For a detailed PCR program and the primers used for amplifying the promoter regions of *CD28*, *PDCD1*, and *TNFSF4*, please refer to ref. [28,29,30].

### 2.2. Promoter–Reporter Assay

#### 2.2.1. Constructing the Template of Promoter–Reporter and Plasmid

To begin, a normal human DNA template was used, and primers with specific restriction enzyme cleavage sites (as shown in Table 1) for the promoter region of the *TNFSF4*, *CD28*, and *PDCD1* genes were designed. After the DNA sequence was confirmed to be correct by sequencing, the promoter fragment was transferred into plasmids using the TOPO TA cloning kit (Invitrogen). Plasmid DNA was then extracted and sequenced in an *E. coli* competent cell (TOP10 or DH5α) to determine the promoter sequence.

#### 2.2.2. Construction of Luciferase Expression Vectors and Transformation

The *TNFSF4/CD28/PDCD1* promoter plasmid with restriction enzyme cleavage sites (SacI and EcoRV for *TNFSF4* and *PDCD1*; SacI and HindIII for *CD28*) and the fluorescent expression vector pNL1.1 [Nluc] (Promega, Madison, WI, USA) were reacted with a restriction enzyme for one hour at 37 °C. Electrophoresis was carried out with 1% agarose gel for 20 min. The pNL1.1 vector with a specific promoter fragment was cut off in the gel, using a PCR/gel purification kit, and the two fragments were then joined with a T4 ligase-connecting enzyme to create a promoter vector with NanoLuc^®^ luciferase expression (Promega). The vector was transformed to competent cells (TOP10 or DH5α). After screening and culture of the successfully transformed cells (LB agar plate), the NanoLuc^®^ luciferase expression vector with the correct promoter fragment was extracted. The promoter report assay used the vector with the correct sequence as the control group, and the plasmid DNA served as the template for site-directed mutagenesis PCR (QuikChange II site-directed Mutagenesis Kit, Stratagene). The report vectors of rs1234314 C > G, rs45454293C > T, rs5839828 C > del, rs36084323 C > T, rs28541784C/T, rs200353921A/T, rs3181096C/T, and rs3181098G/A (Table 1) were constructed in the same way as above.

#### 2.2.3. Dual-Luciferase Reporter Assay

A total of 0.5 μg of the construct promoter–reporter vector, 0.5 μg of the PGL 4.5 [Luc2/TK] vector (Promega, Madison, WI, USA with firefly luciferase used as an internal control to correct transformation efficiency, and Lipofectamine2000 (Invitrogen, Carlsbad, CA, USA) were mixed evenly and allowed to stand for 20 min. They were then co-cultured with 5 × 10^5^ K562 cells in a 24-well plate of RPMI 1640 medium containing 10% fetal bovine serum, 50 U/mL of penicillin, and 50 μg/mL of streptomycin in a CO_2_ incubator for 48 h. The cultured cells were evenly mixed and 80 uL of them were extracted. They were placed in a white flat-bottomed 96-well plate, treated with Promega Passive Lysis Buffer, and allowed to stand for 15 min. The Nano-Glo Dual-Luciferase Reporter with the Dual Luminescent Enzyme Detection Kit Assay System (Promega) was then used for the reaction, followed by luminescence detection with a luminescence enzyme detector (GloMax Discover System, Promega, CA, USA).

### 2.3. Statistical Analysis

The promoter–reporter assay of each SNP mutation was conducted 11–16 times in parallel. We calculated the wild type of each gene by dividing the value of pNL1.1 (NanoLuc) by the value of PGL 4.5 (firefly). The relative light units (RLU) of wild-type constructs were corrected to 1. There was one SNP variation between wild-type constructs and the SNP–reporter construct in order to verify the transcription activity level of the specific SNP. Data beyond plus or minus 2 standard deviations were excluded. The average amount of RLU corresponding to the promoter construct of each SNP was compared using one-way ANOVA in the SPSS 17.0 software package (SPSS Inc., Chicago, IL, USA), and Tukey’s honestly significant difference test was used for post hoc testing. The significance level was set as 0.05.

## 3. Results

### 3.1. TNFSF4 Promoter–Reporter Assay

Both the rs1234314 C > G and rs45454293 C > T reporter assays were tested 16 times in a row. The C-allele of rs1234314 and C-allele of rs45454293 were defined as wild type. It was discovered that rs1234314 C > G had 0.32 ± 0.09 times the RLU of rs1234314 with the C-allele and rs45454293 C > T had 4.630.92 times the RLU of rs45454293 with the C-allele. In ANOVA analysis, the main effect of constructs (F_(2,43)_ = 83.255, *p* < 0.001) was revealed. Following a post hoc analysis, it was discovered that rs1234314 C > G (*p* = 0.003) and rs45454293C > T (*p* < 0.001) have statistical significance with the wild type (Appendix A and Figure 1).

### 3.2. PDCD-1 Promoter–Reporter Assay

Both the rs5839828 G > del and rs36084323 C > T reporter assays were conducted 11 times as independent tests. The del-allele of rs5839828 had 1.37 ± 0.24 times the RLU of the G-allele, and the T-allele of rs36084323 had 0.68 ± 0.07 times the RLU of the C-allele. In ANOVA analysis, it was also shown that there was a main effect between constructs (F_(2,34_) = 70.093, *p* < 0.001). Following a post hoc analysis, it was discovered that rs5839828 G > del (*p* < 0.001) and rs36084323 C > T (*p* < 0.001) have statistical significance with the wild type (Appendix A and Figure 2).

### 3.3. CD28 Promoter–Reporter Assay

In the *CD28* promoter region, four SNPs were examined (rs28541784T > C, rs200353921A > T, rs3181096C > T, and rs3181098G > A), with each promoter–reporter receiving 15 independent tests. It was shown that these SNPs had little effect on transcriptional activity. The RLU value for rs28541784 with the C-allele was 0.92 ± 0.31 times that of rs28541784 with the T-allele, the RLU value of rs200353921 with the T-allele was 0.99 ± 0.18 times that of rs200353921 with the A-allele, and the RLU of rs3181096 with the T-allele was 0.90 ± 0.20 times that of rs3181096 with the C-allele and 0.97 ± 0.25 times that of rs3181098 with the G-allele. In ANOVA analysis, there was no effect between these *CD28* promoter–reporter constructs (F_(4,66)_ = 0.644, *p* = 0.633) (Appendix A and Figure 3).

## 4. Discussion

Co-stimulatory molecules are important in immune regulation and have been linked to the pathogenesis of autoimmune diseases, cancer, and transplant rejection [31]. On antigen-presenting cells, CD28 interacts with CD80/CD86 to provide a second stimulation signal required for T-cell activation [32]. PDCD1 plays a negative regulatory role in the activation process of T cells. It functions as an immune checkpoint and promotes immune tolerance, which can prevent the development of autoimmune diseases; cancer cells can also evade the immune system’s pursuit [33]. In addition, the OX40 ligand encoded by TNFSF4 is the key to coordinating innate and adaptive immune cells and plays an important role in differentiation, activation, inhibition, and apoptosis in the life cycle of immune cells [34].

### 4.1. About TNFSF4 Analysis

Regarding TNFSF4, it was discovered that rs1234314 and rs45454293 of TNSFS4 were associated with post-HSCT GVHD III-IV in acute myeloid leukemia (AML) patients in our previously published HSCT (including bone marrow transplantation and peripheral blood stem cell transplantation) research analysis [29]. Grafts with the rs1234314 C-allele had a 7.39-fold increased risk of GVHD III-IV compared with the G-allele (*p* = 0.011), and the rs45454293 T-allele had a 4.86-fold increased risk of GVHD III-IV compared with the C-allele (*p* = 0.010). In a CBT research study [28], rs1234314 of the TNFSF4 gene was associated with mortality after CBT (*p* < 0.001), and having the CC genotype increased the risk of mortality by 15.4 times. In SLE research analysis [30], the genotype frequency (CC vs. CG vs. GG) of rs1234314 was significantly different between cases and controls (*p* = 0.005). When compared to the CC genotype, the GG genotype was associated with a 4.4-fold increased risk of developing SLE (*p* = 0.004). Analysis of the promoter–reporter assay showed that rs1234314 C > G significantly reduced transcription activity by 0.32 times, while rs45454293 C > T significantly increased activity by 4.63 times.

TNFSF4 expression is known to drive T-cell proliferation, differentiation, and cytokine production [35]. Thus, the higher promoter activity of rs1234314 with the C-allele and rs45454293 with the T-allele was consistent with our previously published findings that suggest that they increase the risk of GVHD III-IV after HSCT in AML patients, as well as the risk of mortality after CBT [28,29]. Furthermore, Tripathi et al. discovered in 2019 that the OX40L(TNFSF4)-OX40 interaction on T cells was linked to the induction and development of acute GVHD (aGVHD) in HSCT, and that treatment with anti-human OX40L mAb could effectively prevent and reduce the severity of aGVHD [36]. Our findings revealed that transplanting the graft with the rs1234314 C-allele and rs45454293 T-allele increased the risk of GVHD III-IV and death because these two alleles had higher transcription activity.

The rs1234314 C to G variation reduced promoter activity by 68% (0.32), which seems to contradict the result for rs1234314 GG where the risk of SLE increased by 4.4 times [30]. However, a Chinese study published in 2017 discovered that the CC genotype of rs1234314 protected against allergic rhinitis in the Chinese Han population [37]. We also found that the risk of SLE in the GG genotype was 4.4 times higher than that in the CC genotype in results of patient analyses. In other words, CC protected against SLE. It has been established that GVHD is caused by the activation of allogeneic T cells in response to recognition of the allogeneic antigen presented by mismatched MHC molecules, resulting in the graft attacking the host [38]. Autoimmune diseases are caused by a loss of immune tolerance that is caused by the overactivation of autologous reactive T cells [39]. Based on this knowledge, we inferred that rs1234314 may play different roles in the prognosis of CBT, HSCT, and autoimmune diseases, which needs to be further verified in the future.

### 4.2. About PDCD1 Analysis

Previous research on the relationship between SNPs and HSCT outcomes [29] found that the C-allele of rs36084323 in the promoter region of the PDCD1 gene in donors was associated with a higher risk of CMV (*p* = 0.0265) and relapse (*p* = 0.0356) in ALL patients, while the G-allele of rs5839828 (*p* = 0.0265) increased the risk of relapse in ALL patients. According to analysis of the effectiveness of CBT [28], when the graft carried at least one T-allele in rs36084323, the CBT case had a 4.2 times greater risk of relapse (95% CI = 1.331–13.320, *p* = 0.012). For SLE [30], it was shown that the T-allele of rs36084323 provided a protective effect.

The PDCD1 promoter–reporter assay revealed that rs5839828 with the del-allele had 1.37 times the transcriptional activity of rs5839828 with the G-allele, whereas rs36084323 C > T decreased transcriptional activity. It is known that the role of PD1 is similar to that of CTLA4, which plays a negative regulatory role in T-cell activation to produce immune tolerance [40]. This regulatory mechanism can help to prevent autoimmune diseases, but it can also keep the immune system from killing cancer cells, resulting in disease relapse [26,41]. Our result showed that rs36084323 T-allele had significantly lower PDCD1 transcriptional activity. As a result, we concluded that transplanting a graft with rs36084323 T-allele may reduce PDCD1 expression, thus reducing T-cell activation and increasing the risk of relapse in ALL patients. In addition, the risk of CMV infection may also increase due to the decrease in T-cell activation. However, SLE patients had a higher frequency of at least one T-allele (CT + TT) in rs36084323 than controls. The possible reason may be the same as the conclusion in the above paragraph: the role of rs36084323 in autoimmune disease and transplantation was not consistent.

Aside from HSCT and CBT outcomes and SLE, rs36084323 has been linked to several cancers and immune abnormalities, including breast cancer, ovarian cancer, esophageal cancer, rheumatoid arthritis, and abortion [42]. A 2014 study showed that non-small-cell lung cancer patients with CC genotype rs36084323 had significantly poorer prognosis [43]. This could be due to rs36084323 SNP variation affecting PDCD1 transcriptional activity. In the reporter assay of rs36084323, we found that rs36084323 C-allele had higher transcription activity than T-allele. In other words, when rs36084323 is associated with the CC genotype, PDCD1 expression is increased. Additionally, PDCD1 is an inhibitory regulator of T-cell activation, which may hurt the elimination of cancer cells.

The effect of rs36084323 G > A (C > T) on transcriptional activity assessed by Ishizaki et al. [44] using a dual-luciferase reporter assay was consistent with ours. Both of our results showed that rs36084323 G > A (C > T) could reduce the transcriptional activity of PDCD1. Furthermore, a 2014 study by Jiao et al. found that rs36084323 with the GG genotype had higher mRNA expression in the PBMCs of SLE patients compared to the AA genotype [45]. Therefore, these reproducible results could prove that our experimental results were correct.

### 4.3. About CD28 Analysis

Previously, we found that rs200353921, rs28541784, rs3181096, and rs3181098 located in the promoter region of CD28 were associated with HSCT effectiveness. AML patients with T-alleles at rs200353921 had a higher risk of relapse (OR = 2.1, 95% CI = 1.06–4.18, *p* = 0.0343). Grafts with the T-allele at rs28541784 increase the risk of chronic GVHD (*p* = 0.0303) in ALL patients (OR = 2.78, 95% CI = 1.11–6.98). The T-allele of rs3181096 and the A-allele of rs3181098 decreased the risk of GVHD in AML patients and ALL patients, respectively. However, in the CD28 gene promoter–reporter assay, these SNP variations were found to not affect transcriptional activity.

GVHD is the most important adverse complication after HSCT. It is an immune system disease that affects many organ systems, including the gastrointestinal tract, liver, skin, and lungs, and has an impact on HSCT outcomes. Research shows that the pathogenesis and severity of GVHD are related to the activation of T cells [46]. CD28-mediated co-stimulatory signals are essential for the initiation and maintenance of T-cell activation. Animal experiments have also found that CD28 is related to the onset and severity of GVHD, and it can even reduce the severity of GVHD by blocking the function of CD28 on T cells [47].

Although there were no statistically significant changes in CD28 SNP and transcriptional activity in this study, we previously discovered that the CD28 SNP rs3181097 G > A could reduce CD28 transcriptional activity and was associated with a significantly lower risk of transfusion reactions [48]. It was proposed that, in addition to these four SNPs, there may be other SNPs in the CD28 gene that regulate gene expression.

### 4.4. The Significant and Further Study

It is well known that correlation does not imply causation. After association analysis, we verified the effect of the SLE- or HSCT-associated SNPs located in the promoter region of genes on transcription activity through reporter assays. Based on the findings, it was determined that several SNPs in one gene had biological functions affecting transcriptional activity (some increasing and others decreasing) and that the final protein level should be the integration result of the co-regulation of these SNPs with biological function. Therefore, it is necessary to test the effect of the haplotype. Next, we will investigate whether the SNPs/haplotypes with a functional effect on transcription activity influence T-cell activation or T-cell differentiation through a cell study model based on the T cells extracted from patients and explore the mechanism of diseases caused by the SNPs/haplotypes through animal models. However, gene expression in vitro does not imply in vivo expression. Thus, the proteins expressed in the serum or on the surface of T cells need to be tested. Additionally, how these SNPs affect transcription activity remains to be clarified, such as exploration through silico studies of the extent to which they alter transcription factor binding sites.

## 5. Conclusions

In summary, the results of this study verify that some promoter SNPs have the biological function of regulating transcriptional activity. As a result, in the future, it will be necessary to thoroughly investigate data related to the genes and proteins involved in autoimmune diseases or outcomes of HSCT to clarify the mechanism of SNPs on diseases. Furthermore, the findings regarding SNP–disease association can only be used to develop a genetic testing kit to aid clinical diagnosis and cannot explain the direct causal relationship between SNPs and diseases.

## Figures and Tables

**Figure 1 jcm-12-02157-f001:**
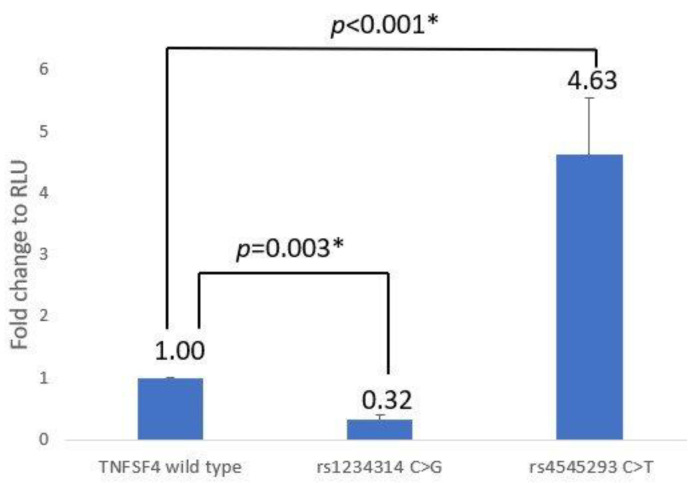
Luciferase reporter assay of the effect of SNPs rs1234314 (C/G) and rs45454293 (C/T) on *TNFSF4* promoter activity. “*” means *p* < 0.05.

**Figure 2 jcm-12-02157-f002:**
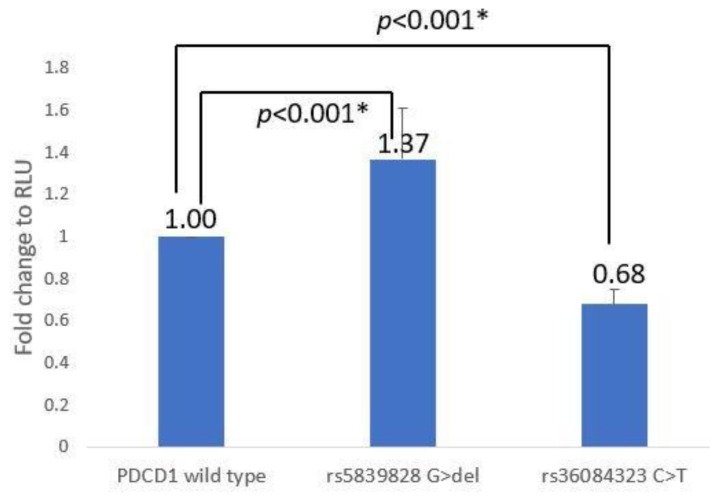
Luciferase reporter assay of the effect of SNPs rs5839828 (G/del) and rs36084323 (C/T) on *PDCD1* promoter activity. “*” means *p* < 0.05.

**Figure 3 jcm-12-02157-f003:**
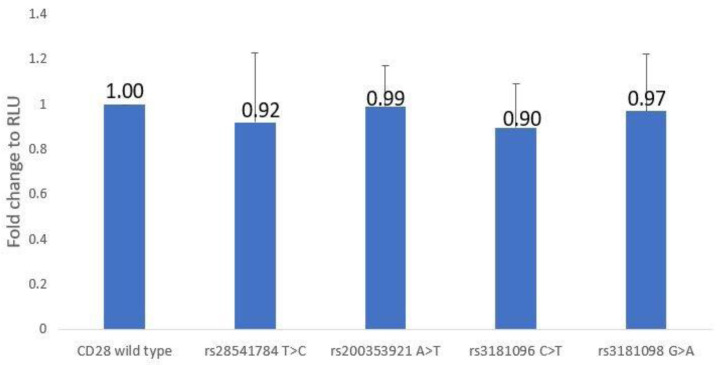
Luciferase reporter assay of the effect of SNPs rs28541784 (T/C), rs200353921 (A/T), rs3181096 (C/T), and rs3181098 (G/A) on *CD28* promoter activity.

**Table 1 jcm-12-02157-t001:** The primer pairs used for site-directed mutagenesis PCR.

Primer	Sequence	NCBI Position
SacI-TNFSF4F	5′-GGCG GAGCTC CT CAA CAC CAG TAT GTT CTC C-3′	173208582
EcoRV-TNFSF4R	5-GGCG GATATC AA TAG GCA AAG GTC CCA GGG C-3′	173207292
Rs1234314CF	5′-TAC ATC ACA TGA G**C**C TGG CAC TGT ACT GGA-3′	173208253
Rs1234314CR	5′-TCC AGT ACA GTG CCA G **G**C TCA TGT GAT GTA	
Rs4545293TF	5′-CTT TCT TTG AGG T**T**G TGG CTG GCC TCA GAA-3′	173208097
Rs4545293TR	5′-TTC TGA GGC CAG CCA C**A**A CCT CAA AGA AAG-3′	
SacI-PD1F	5′-ACTG GAGCTC CA ACC AAC AGT TCT CCA GCC C-3′	241858839
EcoRV-PD1R	5-TTATC GATATC G CCT GGA GCA GCC CCA CCA G-3′	241860275
Rs36084323TF	5′-AAG GGG GAT GGG CC**A** GGA AGG CAG AGG CCA-3′	241859444
Rs36084323TR	5′-TGG CCT CTG CCT TCC **T**GG CCC ATC CCC CTT-3′	
Rs5839828delF	5′-ACCGCCCCAG**CCCCCC**GTCAGGCTGTTGCAGGCAT-3′	241859601
Rs5839828delR	5′-ATGCCTGCAACAGCCTGAC**GGGGGG**CTGGGGCGGT-3′	
SacI-CD28F	5′-TAT GAGCTC AGC AGT TGG CCG TGC TGG TGG AAT-3′	203705243
HindIII-CD28P	5′-TTA T AAGCTT GG GTT CCA GCC CCT CCT CCC CGA-3′	203706675
Rs28541784CF	5′-CCTTCCCTCCCTCCCTCTCTCTTTCTTTCCATCTT-3′	203705806
Rs28541784CR	5′-AAGATGGAAAGAAAGAGAGAGGGAGGGAGGGAAGG-3′	
Rs200353921TF	5′-CCCTCCCTCTTTCTTTCTTTCCTTCTTTCTTTCTTTC-3′	203705818
Rs200353921TR	5′-GAAAGAAAGAAAGAAGGAAAGAAAGAAAGAGGGAGGG-3′	
Rs3181096TF	5′-CTCCTTTTGTGCCCTATTATTTAACCTTGAGGG-3′	203705369
Rs3181096TR	5′-CCCTCAAGGTTAAATAATAGGGCACAAAAGGAG-3′	
Rs3181098AF	5′-GTAACTCCTTTAAACATTTATGCAGATGTTTCCC-3′	203705655
Rs3181098AR	5′-GGGAAACATCTGCATAAATGTTTAAAGGAGTTAC-3′	

NCBI position according to GRCh38.p13. The underlined mutagenesis primer sequence refers to the position recognized by the specific restriction enzyme. The bold sequence refers to the position of site-directed mutagenesis.

## Data Availability

The data presented in this study are openly available in reference number [28,29,30].

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
