# Peer review of "Exploring the Bio-Functional Effect of Single Nucleotide Polymorphisms in the Promoter Region of the TNFSF4, CD28, and PDCD1 Genes"

_jcm, 2023, doi:10.3390/jcm12062157_

Round 1

Reviewer 1 Report

In this article, authors correlate in vitro expression of several mutated promoters with autoimmune diseases. The resulkts are moderately interesant, but it must be taken into account that invotro expression does not imply in vivo expression. The scientfic soundness must be improved with gene expression experiments in tissue portions as biopsies.

Minor issues:

- Gene names must be in italics.

- Line 54: ref 7 is not adequated.

- Line 58: ref 12 is not adequated.

- Line 62: ref 17 is not adequated

- line 91: two points instead of dot.

Line 100: table 1 does not correspond. Idem line 120.

- Line 141: indicate limits of significance.

Table 1: post hoc instead of pot hot. Iden in table 2.

- Fig 1: Meaning of asterics. p value of rs1234314 does not correspond with p in table 1.

- line 181: Discussion is missing.

- Line 228: a cite is needed.

- Suplementary tables are published results, they are not needed.

- Line 252: a space must be deleted.

Author Response

Reviewer1

In this article, authors correlate in vitro expression of several mutated promoters with autoimmune diseases. The resulkts are moderately interesant, but it must be taken into account that invotro expression does not imply in vivo expression. The scientfic soundness must be improved with gene expression experiments in tissue portions as biopsies.

Minor issues:

- Gene names must be in italics.

Response: Thank you for the comment. All gene names are in italics in the manuscript.

- Line 54: ref 7 is not adequated.

Response: Thank you for the suggestion. The ref.7 is edited (line 346-347).

- Line 58: ref 12 is not adequated.

Response: Thank you for the suggestion. The ref.12 is edited (line 354-356).

- Line 62: ref 17 is not adequated

Response: Thank you for the suggestion. The ref.17-19 are edited (line 362-368)

- line 91: two points instead of dot.

Response: The dot (.) after “…SLE” in line 91 is replaced by colon (:).

Line 100: table 1 does not correspond. Idem line 120.

Response: Thanks for your reminder. The correct table 1 was ignored in origin. In present, the table 1 is added in line 105. In addition, the order of other tables is edited at the same time.

- Line 141: indicate limits of significance.

Response: Thank you for the suggestion. The significant level was set as 0.05, which is added in line 143-144.

Table 1: post hoc instead of pot hot. Iden in table 2.

Response: Thanks for your reminder. The “pot” is revised to “post” in the footnote of the original Table 1-3 (they are revised to supplementary Table 1-3).

- Fig 1: Meaning of asterics. p value of rs1234314 does not correspond with p in table 1.

Response: Thank you for the reminder. The p value in the original Table 1 (supplementary Table 1 in revised manuscript) is revised.

- line 181: Discussion is missing.

Response: Thank you for the reminder. The discussion section was misprinted in conclusion section in original manuscript. Now, it is removed to the correct section (line 184-296).

- Line 228: a cite is needed.

Response: In SNP analysis, we calculated the odds ration minor allele of the disease when people with minor allele (lower frequency in the recruited population) compared to those with major allele. It was found that GG genotype had 4.4 times risk of SLE than CC genotype. Thus, CC genotype was protected against SLE. This is a result deduced from our data, so no reference could be cited.

- Suplementary tables are published results, they are not needed.

Response: Thank you for the suggestion. The supplementary tables are deleted.

- Line 252: a space must be deleted.

Response: Thank you for the reminder. The one of the double space before “As a result,…” is deleted in line 251.

Reviewer 2 Report

The authors investigated the effect of several single nucleotide polymorphisms of TNFSF4, CD28 and PDCD1 genes on promoter activity in an in vitro luciferase reporter system. Four of the eight variations were found to be functionally active. 

Major comments: 

Methods: 

1. Six supplementary tables are mentioned in the text, but only four are attached to the article.

2. Supplementary tables should be removed. They contain the results presented in the authors' previous publications in their entirety. In my opinion, the supplementary material should contain new data, as in the original article, otherwise it is sufficient to provide the reference in the text of the article. 

3. According to chapter 2.2.1, Table1 contains the cloning primers, but the table itself is missing. 

4. In chapter 2.2.2, during the preparation of the reporter constructs, the authors incorrectly refer to the introduction of the plasmid into the competent cell as transfection. This process is called transformation. 

5. Table1, referred to repeatedly in chapter 2.2.2, does not contain data corresponding to the text. 

Results: 

6. There is little or no information in the article about the exact location of the investigated polymorphisms within the promoter. A diagram depicting the position of the SNPs examined in relation to the start codon in the given promoter would have greatly helped the understanding. 

7. The tables (Table 1, 2 and 3) in the results are unnecessary, they contain raw data, which is otherwise included in the diagrams anyway, so it is redundant. 

8. The relative luciferase activity measured with a promoterless vector as negative control and/or with a strong promoter containing construct as a positive control is not shown in any of the diagrams (or in the tables). This greatly reduces the evaluability of the measurement, since it is not known whether the inserted DNA sequence really has promoter activity and, if so, to what extent. 

9. The lines in the background of the diagrams are confusing, the labels are blurred with the standard deviation lines. 

10. Why is the average value of the columns shown in Figures 2 and 3, but not in Figure 1? Why is it necessary to enter these data at all, especially if they are already included in the Tables 1, 2 and 3?

Discussion: 

11. The discussion was accidentally moved to the wrong place (to conclusion chapter), it should be moved to chapter 4.  

12. In a very large part, the discussion details the results of the two previous works of the authors of this article, as well as the content of the previously objected Supplementary Tables. 

Author Response

Reviewer2

The authors investigated the effect of several single nucleotide polymorphisms of TNFSF4, CD28 and PDCD1 genes on promoter activity in an in vitro luciferase reporter system. Four of the eight variations were found to be functionally active. 

Major comments: 

Methods: 

  1. Six supplementary tables are mentioned in the text, but only four are attached to the article.

Response: Thank you for the reminder. I had check that there were 4 supplementary tables. And the supplementary tables are deleted according to the following suggestion.

  1. Supplementary tables should be removed. They contain the results presented in the authors' previous publications in their entirety. In my opinion, the supplementary material should contain new data, as in the original article, otherwise it is sufficient to provide the reference in the text of the article. 

Response: Thank you for the reminder. The supplementary tables are deleted according to reviewer’s suggestion.

  1. According to chapter 2.2.1, Table1 contains the cloning primers, but the table itself is missing. 

Response: Thanks for your reminder. The correct table 1 was ignored in origin. In present, the table 1 is added in line 105. In addition, the order of other tables is edited at the same time.

  1. In chapter 2.2.2, during the preparation of the reporter constructs, the authors incorrectly refer to the introduction of the plasmid into the competent cell as transfection. This process is called transformation. 

Response: Thank you for the comment. The “transfection” is revised to “transformation” in line 106 and line 124, and the “transfected” is revised to “transformed” in line 114 and line 115.

  1. Table1, referred to repeatedly in chapter 2.2.2, does not contain data corresponding to the text. 

Response: The Table 1 mentioned in chapter 2.2.2 is same as the Table 1 in chapter 2.2.1. (the added table in this revised manuscript).

Results: 

  1. There is little or no information in the article about the exact location of the investigated polymorphisms within the promoter. A diagram depicting the position of the SNPs examined in relation to the start codon in the given promoter would have greatly helped the understanding. 

Response: Thanks for the comment. The promoter region and the position of the SNPs were shown in Table 1, which is added in the revised manuscript (line 105).

  1. The tables (Table 1, 2 and 3) in the results are unnecessary, they contain raw data, which is otherwise included in the diagrams anyway, so it is redundant. 

Response: Thank you for the suggestion. The original Table 1-3 are revised to supplementary Table 1-3, and the original supplementary Table 1-4 are deleted.

  1. The relative luciferase activity measured with a promoter less vector as negative control and/or with a strong promoter containing construct as a positive control is not shown in any of the diagrams (or in the tables). This greatly reduces the evaluability of the measurement, since it is not known whether the inserted DNA sequence really has promoter activity and, if so, to what extent. 

Response: The PGL 4.5 [Luc2/TK] vector (Promega) with firefly luciferase used as an internal control to correct transformation efficiency, which was shown in line 123-125. The wild type construct was used as positive control. There was one SNP variation between wild type construct and SNP-reporter construct, in order to verify the transcription activity level of the specific SNP. In addition, the RLU of wild type construct was corrected to 1 to compare with the RLU of other constructs, which is explained in line 137-139.

  1. The lines in the background of the diagrams are confusing, the labels are blurred with the standard deviation lines. 

Response: Thank you for the suggestion. The lines in the background of the diagrams are removed.

  1. Why is the average value of the columns shown in Figures 2 and 3, but not in Figure 1? Why is it necessary to enter these data at all, especially if they are already included in the Tables 1, 2 and 3?

Response: Thank you for the suggestion. The average value in Figure 1 is added. Table 1-3 has been removed according to reviewer's comment, so average value is necessary to show in the diagram.

Discussion: 

  1. The discussion was accidentally moved to the wrong place (to conclusion chapter), it should be moved to chapter 4.  

Response: Thanks for your reminder. The discussion section is moved to chapter 4 (line 184-296).

  1. In a very large part, the discussion details the results of the two previous works of the authors of this article, as well as the content of the previously objected Supplementary Tables. 

Response: Because we further conducted functional analysis based on previous research results to verify that the correlation of SNPs to disease development has biological functional significance, the previous results would be discussed together with the results of functional analysis.

Round 2

Reviewer 1 Report

Authors do not answer to this issue: The scientfic soundness must be improved with gene expression experiments in tissue portions as biopsies.

Without these experiemnts I think this paper is not adequate for JCM.

Author Response

Reviewer1

Authors do not answer to this issue: The scientfic soundness must be improved with gene expression experiments in tissue portions as biopsies.

Without these experiemnts I think this paper is not adequate for JCM.

Response: Please accept my apologies for ignoring the issue. In this study, the SNPs associated with HSCT and SLE were verified that they had functional effect on transcription activity through reporter assay in vitro. Indeed, this is a good suggestion. We know that the gene expression in vitro does not imply in vivo expression, and we will work hard in this direction in the future. This explanation has been added in line 305-307 of revised manuscript.

Reviewer 2 Report

The authors investigated the effect of several single nucleotide polymorphisms of TNFSF4, CD28 and PDCD1 genes on promoter activity in an in vitro luciferase system. Four of the eight variations were found to be functionally active. The authors made significant improvements to their manuscript during the review process. 

Major comments: 

1. I strongly recommend that the authors supplement the experimental results with at least in silico studies. This means that it would be worthwhile to analyze the effect of the four polymorphisms (rs1234314 C > G, rs45454293 C > T, rs5839828 G > del, rs36084323 C > T) of the two genes affecting the promoter activity (TNFSF4, PDCD-1) on the transcription factor binding site with suitable TF binding site search programs (e.g., Jaspar, Transfac). What transcription factors can bind to the promoter regions in question? Can the given polymorphism affect their binding? To what extent? What are the consequences?

Author Response

Reviewer2

The authors investigated the effect of several single nucleotide polymorphisms of TNFSF4, CD28 and PDCD1 genes on promoter activity in an in vitro luciferase system. Four of the eight variations were found to be functionally active. The authors made significant improvements to their manuscript during the review process. 

Major comments: 

  1. I strongly recommend that the authors supplement the experimental results with at least in silico studies. This means that it would be worthwhile to analyze the effect of the four polymorphisms (rs1234314 C > G, rs45454293 C > T, rs5839828 G > del, rs36084323 C > T) of the two genes affecting the promoter activity (TNFSF4, PDCD-1) on the transcription factor binding site with suitable TF binding site search programs (e.g., Jaspar, Transfac). What transcription factors can bind to the promoter regions in question? Can the given polymorphism affect their binding? To what extent? What are the consequences?

Response: Thanks for reviewer. Because we are not yet fully familiar with silico studies, we tried to find out whether these four SNPs affect the binding site activity of transcription factors from other literatures. After thoroughly searching, we found nothing. Indeed, this is a good suggestion, and we will work hard in this direction in the future. In addition, “how does these SNPs affect the transcription activity remains to be clarified, such as altering the transcription factor binding site through silico studies” has been added in line 307-309.
